# Best Practices and Progress in Precision-Cut Liver Slice Cultures

**DOI:** 10.3390/ijms22137137

**Published:** 2021-07-01

**Authors:** Liza Dewyse, Hendrik Reynaert, Leo A. van Grunsven

**Affiliations:** 1Liver Cell Biology Research Group, Department of Basic Biomedical Sciences, Vrije Universiteit Brussel, 1090 Brussels, Belgium; liza.dewyse@vub.be; 2Department of Gastro-Enterology and Hepatology, Universitair Ziekenhuis Brussel, 1090 Brussels, Belgium; Hendrik.Reynaert@uzbrussel.be

**Keywords:** PCLS, chronic liver disease, 3D, in vitro liver

## Abstract

Thirty-five years ago, precision-cut liver slices (PCLS) were described as a promising tool and were expected to become the standard in vitro model to study liver disease as they tick off all characteristics of a good in vitro model. In contrast to most in vitro models, PCLS retain the complex 3D liver structures found in vivo, including cell–cell and cell–matrix interactions, and therefore should constitute the most reliable tool to model and to investigate pathways underlying chronic liver disease in vitro. Nevertheless, the biggest disadvantage of the model is the initiation of a procedure-induced fibrotic response. In this review, we describe the parameters and potential of PCLS cultures and discuss whether the initially described limitations and pitfalls have been overcome. We summarize the latest advances in PCLS research and critically evaluate PCLS use and progress since its invention in 1985.

## 1. Introduction

In 1980, Carlos Krumdieck was the first to describe the Krumdieck tissue slicer that allowed the fast production of thin tissue slices [1]. Shortly after, in 1985, precision-cut liver slices (PCLS) were described for the first time by Smith et al. as an in vitro model for toxicity testing [2]. The principle was simple; punched cylindrical cores of liver tissue were subsequently cut into 250 µm slices that could be cultured in regular tissue culture plates. Scientists were convinced of the benefits of this culture model, especially since the hepatocyte cultures at that time lacked liver cell heterogeneity and thus would not sufficiently mimic the in vivo liver. Moreover, hepatocytes deteriorated rapidly once brought in culture [3]. Initially, PCLS cultures were used for metabolic studies and toxicity testing [4,5,6] but from 2005 onwards, the focus of PCLS experiments shifted from toxicity testing towards studies of chronic liver disease, such as fibrosis [7].

In 2010, de Graaf et al. published an extensive protocol on the production of human and rat tissue slices (liver and intestine) [8], which is still the reference method for researchers working with PCLS cultures. In short, preceding the PCLS preparation, animal liver tissue is harvested or human donor tissue is obtained. Subsequently, the liver tissue is cut with a specialized slicer. Depending on equipment or techniques used, cylindrical cores are made before or after slicing of the tissue. Although PCLS cultures offer a lot of opportunities as an in vitro model of chronic liver disease, the procedure of making PCLS is associated with at least one main disadvantage as well; the slicing of the tissue induces liver damage and cell death, which eventually triggers a repair and regenerative response, resulting in fibrosis once the slices are put in culture. Although not widely studied in liver slices, damage and death of hepatocytes leads to differentiation of the hepatic stellate cells (HSCs) into an activated myofibroblast-like cell, responsible for the excessive production of extracellular matrix (fibrosis). Due to the slicing of the liver tissue during PCLS production, HSC activation will occur within the first 48 h of culture [9]. Healthy slices’ cultures are thus limited to a short culture life span.

Several reviews discuss the applicability of PCLS. The most recent, by Othman et al. lists the latest advances in PCLS research and the applicability of PCLS cultures [10]. Palma et al. give an overview of the applicability of human liver slices to study different liver diseases [11], while Godoy et al. compared PCLS culture conditions to different in vitro culture systems for hepatotoxicity testing [12]. This review will focus on the evolution of PCLS parameters and the read-outs used for PCLS analysis.

## 2. Precision-Cut Liver Slice Parameters

As in every experimental technique, many parameters should be considered when setting up PCLS or slice cultures, including equipment, slice thickness and diameter as well as culture medium. For this review, we analyzed each of these parameters in over a hundred studies that used liver slice cultures in the past 10 years and briefly discuss our findings. For a detailed table and reference list of all papers included in the analysis, please see the Appendix A and enclosed reference list (Appendix A).

### 2.1. Tissue Slicer

The most widely used slicer is still the Krumdieck (now ‘Alabama’) tissue slicer, as more than 50% of all PCLS-research papers use this instrument to prepare liver slices (Figure 1). Although the Brendel/Vitron slicer (9.35%) is another ‘standard’ slicer, since its introduction in 2009, the Leica VT1200S vibrating blade microtome (vibratome) has gained popularity (19.63%) [13]. It is hard to compare all slice instruments, as each of them has its own advantages and disadvantages. The main difference is when using the Krumdieck tissue slicer, cylindrical cores have to be made before slicing, while for the vibratome, cylindrical punches can be made after slices are obtained. Cylindrical cores can be made beforehand for the vibratome as well, however these cores then need agarose embedding. Finally, the Krumdieck tissue slicer might be slightly faster than the vibratome, but handling of a vibratome is easier and the thickness of the slice can be tuned to micrometer level. Finally, the Leica vibratome is slightly less expensive than the Krumdieck tissue slicer.

Only a few studies have compared the different instruments. Price et al. showed no significant differences in rat slice viability and protein content over a culture period of 72 h using either a Krumdieck tissue slicer or a Brendel tissue slicer [14]. Zimmermann et al. compared liver slices from five different species made with the Leica VT1200S vibratome or with the Krumdieck tissue slicer. They demonstrated a higher accuracy and reproducibility for rat, mouse, and human tissue slices using the vibratome. Slices showed a higher viability (ATP content) over a 72 h culture period when made by the vibratome compared to the Krumdieck tissue slicer, suggesting a higher survival rate for vibratome-produced slices [13]. While these studies compared instruments, they did not take into account that perhaps downstream handling should be different for the different instruments. Therefore, it is difficult to prioritize the use of one machine of the other, as it will also depend on availability of the equipment in your institute. Nevertheless, while the techniques to make slices seem quite simple; hands-on experience is probably as important as the choice of the equipment.

### 2.2. Liver Slice Thickness

The commonly used thickness of PCLS is 250 µm, as this allows complete and full access and supply of nutrients and oxygen to the inner cell layers, which would not be the case when slices would be thicker than 300 µm [15]. However, slices can be made as thin as 100 µm if necessary [16]. Slice thickness is important to assure good quality, as thinner slices would result in low viability rates, as the ratio of damaged to undamaged cells is too big. A large majority of all investigated papers use slices with a thickness within the range of 200–300 µm (85.19%), while only 5.56% used slices thinner than 200 µm and 4.63% thicker than 300 µm (Figure 1). This shows that the majority of the PCLS studies use slices that are within the recommended thickness range and thus guarantee optimal cell viability.

### 2.3. Liver Slice Diameter

The Krumdieck tissue slicer is still the preferred instrument to prepare PCLS. It allows the preparation of cylindrical tissue cores with a diameter of 3, 5, 8, 10, or 15 mm, which are subsequently sliced. The optimal size of the core used depends on the amount of tissue available on the one hand and the requirements of the experiment on the other hand. Using a vibratome is less limiting regarding slice diameter, as tissue sections can be punched afterwards by a biopsy puncher, which is commercially available in the desired size range of 1 mm to 12 mm (with 0.5 mm intervals). With the exception of a 25-year-old paper, published by Fisher et al., where they compare 6 mm and 8 mm liver slices [17], we could not find any research papers that compared the viability and reproducibility of liver slices with different diameters. Currently, irrespective of equipment, 5 mm (32.41%) and 8 mm (37.04%) liver slices are the most widely used diameters to carry out liver slice cultures (Figure 1). The use of 5 and 8 mm wide slices is most likely based on the first slice studies and the methods paper of de Graaf et al. [8]. In addition, this diameter also guarantees easy handling, a high number of slices from one liver and it fits with the dimensions of the well plates. To date, there is no clear proof to justify the use of 5 or 8 mm slices over 3 or 10 mm slices. However, it is tempting to consider downscaling the size of the slices since it would make high throughput screenings and incorporation into microfluidic devices more feasible.

### 2.4. Liver Slice Culture Conditions

According to de Graaf et al., the recommended volume-to-slice ratio for medium is 0.25–0.30 milliliter per milligram of tissue [8]. In practice, 8 mm slices with a 250 µm thickness are mostly cultured in a 12-well plate with 1.3 mL medium. PCLS are ideally refreshed at least once per 24 h and medium should be moved by shaking [8,18] (e.g., rocking platform), and preferably at 90 rpm. Additionally, liver slices are most often cultured with William’s’ E medium (WME medium) in regular well plates with the well-plate format being dependent on the chosen PCLS diameter. The majority (89.81%) of the PCLS papers up to date (Figure 1), only culture the slices for a maximum of a 72 h. However, several papers describe attempts to prolong PCLS culture and slice viability by adding other medium supplements or by changing the culture set-up. Szalowska et al. demonstrated that changes in culture medium (glucose, insulin, serum, medium volume) did not have a significant effect on liver slices cultured for 24 h. Changes in oxygen concentration (20%, 40%, 60%, 80%), on the other hand, showed the highest viability of the PCLS with the highest oxygen concentration [19], confirming that the standard 95% oxygen supply is required to maintain the highest viability rates [20]. Koch et al. showed that culturing PCLS in a liquid/liquid culture system, consisting of WME medium (upper phase) and non-water soluble PFD (perfluorodecalin, an oxygen carrier, lower phase) preserves ATP levels better and prevents hepatocyte necrosis [21]. In 2015, Starokozhko et al. compared regular WME medium with or without the addition of RegeneMed medium additives during a 5-day culture period of rat PCLS. Rat slices cultured in the supplemented medium did not necessarily result in an improved viability or functionality [22]. Only 2 years later, the same group reported the culture of human slices for 5 days in WME medium with or without RegeneMed or Cellartis Hepatocyte Maintenance medium additives. They showed that human slices cultured in supplemented medium better preserved viability, protein content, glucose homeostasis, and albumin synthesis than the slices cultured in regular non-supplemented medium [23]. A recent study used improved PCLS culture conditions that prolonged rat and human PCLS’s life span for at least 6 days by making use of a bioreactor set-up [24]. In contrast, Wu et al. cultured human PCLS for at least 15 days [25] without profound changes to the culture protocol. They showed that human PCLS cultures stabilize after 4 days, with a sustained viability from day 7 until day 15 (MTS activity). In 2021, studies showed that human PCLS could be kept in culture for 21 days in WME medium in inserts with maintenance of cell viability up to 55% at end of the culture [26]. We note that both these long-term PCLS culture achievements have used slices of human origin, so far no papers describe similar long-term studies on PCLS from other (murine) origin.

In order to improve liver slice culture longevity, parameters used in tissue slice research of other organs such as brain, kidney, heart, lung, or intestine could perhaps be extrapolated to liver slice cultures. Unfortunately, most other tissue slice cultures have to deal with a limited lifespan as well. Heart slices are limited to a lifespan of 24 h [27,28], kidney slices to a lifespan of 72 h [29,30], while lung slices can be cultured, similar to PCLS, for 1 week [31] to 10 days [32]. In contrast, it has been shown that brain slices obtained from post-mortem resected human brain tissue can be cultured for up to 30 days, with for some donors even an exceptional lifespan of 78 days, while neuronal morphology is preserved [33]. The culture conditions of these brain slices however do not show significant differences to regular brain cultures. This suggests that a short lifespan is intrinsic to precision-cut tissue slices in general, although the lifespan might be increased by further optimization of culture conditions or might just be dependent on the tissue type.

### 2.5. Species

Until now, PCLS cultures are mainly established from murine or human livers. The majority of PCLS is made of human (29.75%), followed by rat (28.93%) and mouse origin (24.79%, Figure 1). Of course, the use of human tissue is highly desirable and recommendable for their relevance to human liver toxicity and disease. However, human donor tissue is scarce and more importantly, the tissue quality is highly variable which strongly affects the reproducibility of the results [34]. For example, Jetten et al. have shown that the variability in basal CYP enzyme levels in human liver slices produced of different human donors could be reaching up to more than a 500-fold difference [35]. These differences in basal CYP enzyme levels resulted in a 400-fold inter-individual difference for APAP-induced cytotoxicity and a 6-fold difference for APAP-induced genotoxicity. These variances are expected to be lower in PCLS experiments conducted from lab animal sources, however no such study has been done yet. Our literature search shows that murine tissue (rat and mouse) is more often used as a source for liver slices (53.9%) than any other species such as bovine, fish, hamster, and porcine (16.53%, Figure 1). The preferred species of use is depending on the resources and the investigated disease, as inter-species differences, as well as differences in gender, can influence susceptibility to and can change disease outcome of liver disease [36].

### 2.6. Disease Model

The use of PCLS was initially described as a model for toxicity testing [2]. PCLS were mainly used to study the metabolization of compounds or to investigate the toxic effects of drugs and compounds with the focus on hepatocyte cell death, while other liver cell types were neglected. However, since 2005, the PCLS model has been used to study fibrosis and to investigate the pro- [37,38] and anti-fibrotic [7,39,40] properties of compounds. As the production of the slices is associated with a spontaneous and culture-induced fibrotic response, the anti-fibrotic characteristics of compounds can easily be investigated. However, it should be kept in mind that this ‘artificial’ induced fibrotic response might not completely resemble the in vivo initiated and observed fibrotic processes due to drugs, alcohol or the metabolic syndrome.

Testing pro-fibrotic compounds, on the other hand, remains more difficult as the initial fibrotic stimulus introduced by the cutting of the liver (hepatocyte damage) and subsequent culture of the slices needs to be surpassed by potential pro-fibrotic substance. Even so, compounds such as TGF-β1 or PDGF-ββ can easily stimulate fibrosis, or more precisely hepatic stellate cell (HSC) activation [24], as both factors directly act on these HSCs. However, indirect stimuli through, e.g., hepatocyte-damage or fat accumulation are more challenging as the slicing procedure and culture is accompanied by the loss of hepatocyte cytochrome 450 enzymes, which are necessary for compound metabolization. This is emphasized by the fact that, so far, only two research papers, published by van de Bovenkamp et al. [38] and by Vatakuti et al. [37] demonstrate the possibility of inducing a fibrotic response (upregulation HSC activation markers on mRNA) in a PCLS culture by making use of CCl_4_ (hepatotoxin), although hepatocyte-damage is not proven or thoroughly investigated in these papers. Only recently, Kartasheva-Ebertz et al. [26] demonstrated that fibrosis-associated markers (TGF-β1 gene expression, secretion, and protein level) could be upregulated in human F0-F1 slices, even 1 day after inoculation and infection with the Hepatitis C virus. Moreover, they describe the induction of pro-fibrotic features in infected and non-infected human liver slices (e.g., increase in TGF-β1 and pro-col1a1 expression) after exposure to ethanol (alcoholic liver disease) or palmitate (lipotoxicity). However, no significant changes were observed in viability in these human liver slices after treatment, so the effects of the infection or the palmitate or ethanol treatment on hepatocyte damage are not clarified.

In addition to toxicity testing/metabolization (32.52%) and liver fibrosis (21.14%, Figure 1), more recently, slice cultures have been used to model other liver diseases as well. As the processes and pathways underlying and resulting in chronic liver disease are hard to mimic faithfully in vitro in slice cultures, an alternative approach could be to establish these chronic liver diseases in vivo in mice or rat and to subsequently make slices of the diseased livers. This approach has emerged in several PCLS research papers the past few years, not only to model fibrosis (2 weeks CCl4 treatment) [24] or NAFLD (12 weeks CDAA diet [41], diet-induced obesity [42]) but also acute-on-chronic liver disease (susceptibility of fat PCLS to ethanol exposure [42], or effect of hepatitis C infection on fibrotic PCLS [26]). However, it should be kept in mind that in this way, the experiments’ outcome could differ from the in vivo disease, as little is known about the influence of the slice culture-induced fibrotic response on the progression of the disease phenotype in vitro. The possibility exists that this fibrotic response takes the upper hand.

## 3. Precision-Cut Liver Slice Read-Outs

One of the main advantages of PCLS cultures is that multiple read-outs can be obtained of one single slice, as both the slice itself and the culture medium can be analyzed. Although liver tissue allows the use of a wide range of analysis methods, only a few techniques are commonly used to validate and analyze PCLS cultures.

The most frequently used analysis techniques, divided in slice/tissue or medium analysis, will be discussed in the following paragraph. The reasons and limitations why other established common research read-out are rarely used to analyze PCLS will be discussed as well. An overview of the frequently used techniques for PCLS analysis, including cost and (dis)advantages is given in Table 1.

### 3.1. Culture Assessment

#### 3.1.1. Viability and Cytotoxicity

In their methods paper, de Graaf et al. recommend monitoring adenosine triphosphate (ATP) content and slice morphology as a viability control for each experiment [8]. Indeed, of all research papers using a viability or cytotoxicity read-out (65.7% of investigated papers), the quantification of ATP content is the most used assay (47.5%) to determine slice viability, followed by the lactate dehydrogenase (LDH) leakage assay (23.2%). To a lesser degree, an MTT- (3-(4,5-dimethylthiazol-2-yl)-2,5-diphenyl-2H-tetrazolium bromide) or MTS-assay(-(4,5-dimethylthiazol-2-yl)-5-(3-carboxymethoxyphenyl)-2-(4-sulfophenyl)-2H-tetrazolium) (8.1%) or protein content quantification (10.1%) are used. Other viability assays include staining with PI or Hoechst and quantification of albumin or potassium (K^+^) retention.

Both the ATP assay as the LDH leakage assay are methods that can generate a measurable signal in a short period of time. While the amount of ATP measured can be correlated to the number of cells that produce energy in slice cultures [43] (viability assay), the release of LDH in the culture medium is proportional to cell death [44] (cytotoxicity assay). Both factors, are thus indicators of slice viability, or at least give an estimation. Studies have shown similar results when comparing ATP content with LDH leakage, as both techniques resulted in almost equal IC50 values, when testing tamoxifen in HepG2 cells [45]. Moreover, Kartasheva-Ebertz et al. analyzed ATP production and LDH release of cultured human liver slices over time and found similar results [26], proving the assays’ resemblance in liver slice studies as well.

Both techniques are quite easy and straightforward; however, the ATP assay has two main disadvantages compared to the LDH assay. The ATP assay requires complete lysis of PCLS, requiring more steps and sonication of the slices or more specialized lysis buffers [8], while analysis of LDH leakage only requires the collection of culture medium. Moreover, the ATP assay only allows end-point analysis, while the LDH release in the culture medium allows the follow-up over culture time.


ijms-22-07137-t001_Table 1Table 1Overview of read-outs used for PCLS analysis. For each enlisted technique the PCLS-related advantages, disadvantages or difficulties, analysis time (including sample preparation), and an indication of the cost per data point are given. Price range is from several euros/sample (€) to thousand euros/sample (€€€€).Read-OutAdvantagesDisadvantages/DifficultiesAnalysis TimeCost per Data PointViability and CytotoxicityLDH leakageSlice specific follow-up over timeHigh throughputReader required±1 h€ATP assayHigh throughputEnd point analysis onlyTotal tissue lysis requiredReader required±1 h€MorphologyEvaluation of liver structureInterpretation not always straightforwardEmbedding technically challenging±3 h€Slice analysisqRT-PCRm(i)RNA analysisEvaluation of multiple m(i)RNA’sSelective m(i)RNA measurements±3 h€€Gene expression profilingGeneration of unbiased and complete mRNA profileNeed for bioinformatic skillsNeed for (nucleomics) infrastructureSeveral days€€€€Western BlotMedium sensitivityHigh specificityLow throughputHigh tissue input required1 day€€€Hydroxyproline assayQuantitativeHigh tissue input required Accurate measurement of input1 day€€Immunohistochemical stainingsCan provide more detailed insight into zonation and cell type specific protein expressionEmbedding technically challengingRelative low throughputLimited amount of stainings per slice>1 day€€Triglyceride quantificationFatty/metabolic liver disease specificTotal homogenization of tissue required±2 h€€GSH contentFunctionalAffected by fat storage [46]±1 h€€Medium analysisALT, AST measurementsStraightforwardDirect indicator of liver damageConcentration of medium might be required/sufficient level of damage required to exceed thresholdLow throughputMinutes€ELISAQuantitative Multiple proteins can be tested for 1 sliceLow throughputHours€€€Protein ProfilingGeneration of unbiased and complete protein profileNeed for bioinformatic skillsNeed for (proteomics) infrastructureSeveral days€€€€


#### 3.1.2. Morphology

Not only the quantification of viability can indicate the health status of the cultured PCLS; slice morphology reveals information about culture quality. Light microscopy can give a first insight into tissue wellbeing. Damaged or injured PCLS will become darker in color. Stainings, such as the most widely used hematoxylin and eosin (H&E) staining or a PAS staining, can give more insights into the histology and quality of the liver tissue but only allows for end-point analysis. However, interpretation of these stainings often requires a profound knowledge of liver (patho)histology, as non-experts may overlook minor differences.

### 3.2. Slice Analysis

#### 3.2.1. RNA Analysis

Real-Time Quantitative Reverse Transcription PCR—mRNA analysis is the mostly used method to analyze cultured PCLS. The RNA of PCLS can be obtained by homogenization of the slice tissue, which can be done by making use of a homogenizer, a tissue crusher or a similar instrument. This step is usually followed by a Trizol RNA extraction. Subsequently, mRNA is reverse transcribed and analyzed by qRT-PCR. Although a sufficient RNA yield can be obtained of only one slice, most researchers pool PCLS for mRNA analysis. The observation of big variances in tissue input for PCLS analysis emphasizes the importance of the use of good reference genes.

Gene expression profiling—Gene expression profiling using arrays allows for the detection of thousands of genes at once. Using gene expression profile analysis one can identify modified pathways or processes that can give more insight into the mechanisms leading to the hepatotoxicity and/or subsequent fibrosis occurring in the slice. Bigaeva et al., for example, investigated the differences and similarities between different mouse organs in basal ECM profiles and their culture-induced fibrotic signatures making use of a gene panel of ECM related genes [47]. Beaumont et al. performed transcriptome profiling to investigate the effect of indole on murine liver slices prepared of genetically obese (ob/ob) mice [48]. Microarray analysis has also been performed in human slices by Ijssennagger et al. to investigate the effect of obeticholic acid [49]. RNA sequencing (RNA-seq) provides a more precise insight and measurement of transcriptomic levels and changes in gene expression. Although RNA-seq is nowadays widely implemented, up to now the technique is still not frequently applied to PCLS cultures. Most PCLS research to which RNA-seq is applied, have been carried out in fish-derived liver slices in which the effect of culture time is compared to conventional 2D hepatocyte cultures and total liver [50] or the effect of compounds [51,52]. Bigaeva et al. performed RNA-seq of healthy and diseased murine and human tissue slices in order to investigate the differences in species and organs [53]. Hoowever, Kenerson et al. performed RNA-seq on human tumor slices and investigated the in vitro cancer slice properties to in vivo growth [54].

#### 3.2.2. Protein Analysis

Western Blot—Protein analysis is often used in parallel to gene expression analysis, as changes at the transcriptional level do not necessarily translate into changes at the protein level (to the same extent) and vice-versa. Protein analysis of the slices itself, nevertheless, is not common in PCLS studies, as only 28% (30/107 papers) of the analyzed PCLS papers perform protein analysis. An easy explanation for this is that most researchers pool several slices to obtain enough protein to run a Western blot analysis and this requires many slices if one wants to carry out time-course experiment or test multiple conditions.

Hydroxyproline assay—The hydroxyproline assay is by far one of the most quantitative techniques for determining collagen content, since it quantifies hydroxyproline concentrations, which is a component of collagen and makes this assay interesting in the context of fibrosis-related studies. However, the main limitation of this technique is that a lot of input (tissue) is required to reach the detection limit of the assay. For this, weighing of the tissue is mandatory, which is not easy for PCLS to get accurate results. Moreover, the requirement of pooling multiple PCLS to obtain results for a certain read-out discards the advantage of making use of the PCLS system, as one of the main advantages of the system is that one condition can be modelled in one single slice, of which you can obtain multiple read-outs. Probably, for these above-mentioned reasons, only two PCLS research papers were found where this technique is applied to the liver slices, published by Paish et al. (2 × 8 mm discs) [24] and Westra et al. [55] (3 × 5 mm discs).

Immunohistochemical stainings—Most immunostainings performed on PCLS are done on paraffin-embedded sections of the liver slices. Next to H&E stainings, mostly a Picrosirius Red (PiSR) staining or to a lesser extent, a DAB (3,3′-Diaminobenzidine) staining are performed. A few papers also make use of cryosections for immunofluorescence (IF) [56,57] or Oil Red O (ORO, lipid accumulation) stainings [22,58] and some studies do not even section the slices for the ORO staining [59,60]. Overall, only a few PCLS research papers (<10% of investigated papers) show immunostainings, with HSC markers (e.g., αSMA, Collagen) and Ki67 being the mostly used antigens.

For paraffin embedding, the biggest hurdle to overcome, especially when working with smaller diameter discs, is the handling and manipulation of the liver slices. PCLS are hard to hold in place while embedding, and moreover, the slices shrink and get folded easily, which makes it hard to prepare sections of good quality. For example, folding of the slice may result in sections containing only the slice border or center. However, once this handling is mastered, sections can be easily made, which allows for instance serial stainings to evaluate differences between the center of the slice and the outer layers. Moreover, multiple stainings can be done on only one PCLS, acquiring even more insight if iterative staining technologies are implemented on one section using for instance the MILAN protocol [61,62] or Miltenyi Biotec’s MICS technology [63].

An alternative to overcome the embedding step would be to stain the whole slice, which is relatively easy and can be done in-well. However, more specialized microscopes might be required for good imaging of these stainings, such as confocal- or light sheet microscopes. Moreover, the amount of stainings that can be performed per slice is limited in a normal setting, although this could also be overcome using MILAN or even MICS technologies. Finally, PCLS are not often analyzed using electron microscopy [64,65].

#### 3.2.3. Other Tissue Analyses

Triglyceride quantification—Following homogenization, the amount of intracellular triglyceride content in PCLS can be quantified [21,42,66]. Although this analysis is not frequently used, it can reveal information about intracellular fat storage, and thus an interesting parameter for research concerning fatty liver disease.

Glutathione (GSH)—The liver plays a major role in GSH production and homeostasis in the human body. GSH is involved in many vital functions such as fibrogenesis and detoxification mechanisms as it functions as conjugator, and is therefore involved in liver pathologies [67]. While being mainly produced by the hepatocytes, GSH and (total) oxidized GSH (GSSG) levels can be measured in PCLS tissue by spectrophotometry [68,69].

### 3.3. Medium Analysis

Multiple analyses can be done on PCLS medium, as this is excessively present and for most assays, only a relatively small volume is required. Detection of the many factors that liver slices secrete and produce reveals a lot about the processes and pathways in the slice itself. Moreover, the main advantage is that analysis of the culture medium does not require further slice processing or tissue lysis.

ALT and AST measurements—ALT (alanine aminotransferase) and AST (aspartate aminotransferase), are both serum markers of liver injury [70] as they are released into the circulation upon hepatocyte membrane damage. In PCLS culture medium, these enzymes can be used as an indicator of hepatocyte death as well, although only a few papers describe this technique [24,48,71,72,73]. Due to the relatively high volume-to slice ratio, ALT/AST levels are difficult to detect, while the levels at the first day are relatively high during the first 24 h of culture due to the slicing of the liver. These aspects might inhibit researchers to use this relatively straightforward analysis since all larger hospitals have a service that can measure these liver parameters or it can be done using simple devices such as a Spotchem EZ.

ELISA—Secreted proteins can be quantified with an enzyme-linked immunosorbent assay (ELISA). Most research papers applying ELISAs on slice culture medium measure proteins secreted by the hepatocytes (albumin [24,73,74]) or inflammation related factors (IL-6 [42,75], TNFα [42,76], CXCL, and IL-1β [77]). Only one paper describes the detection of pro-collagen1α1 [41], which is a HSC-specific secreted factor. Overall, for unexplainable reason, ELISA is not generally applied to PCLS research.

Protein profiling—Although protein profiling, just like RNA sequencing, can give a more detailed insight on ongoing pathways and processes, this technique is not often applied to PCLS cultures. Hadi et al. performed protein/metabolite profiling using matrix-assisted laser desorption–ionization—time of flight mass spectrometry (MALDI-TOF MS) to identify the similarities and differences between rat and mouse PCLS exposed to APAP and AMAP during 24 h and to extrapolate this knowledge to human [78]. In a follow-up study, van Swelm et al. applied the same technique (MALDI-TOF MS) to the culture medium of rat, mouse and human PCLS and compared this data to urinary protein profiles. They observed similar toxicity-induced changes in the urinary and medium protein profiles for mouse, suggesting that PCLS cultures can be useful for biomarker detection [79].

miRNA analysis—MicroRNAs (miRNAs) are small single-stranded non-coding RNA molecules and can modulate gene expression post-transcriptionally. This way, miRNAs can regulate physiological functions of the liver and have been linked with liver pathogenesis [80,81,82]. Analysis of (secreted) miRNAs can therefore give further insights on liver disease initiation or progression. However, no PCLS research papers were found describing the analysis of miRNAs secreted in the culture medium of PCLS, although Zarybnicky et al. investigated the inter-individual variability of tissue expressed miRNAs in human PCLS exposed to APAP [83].

We have listed the most appropriate read-outs to consider for the analysis of PCLS cultures depending on the application of the slices in Table 2. These can be considered as the minimal analysis one should carry out for the characterization of the PCLS cultures. For further in depth analysis, bulk-, single-cell, nuclear-, or even spatial RNA sequencing can be considered, as well as other omics approaches [79,84,85].

### 3.4. Other Applications

Dissociation for individual cell evaluation—The dissociation of cultured PCLS allows the analysis on a single-cell type level and could give valuable insights into cell-type-specific changes, disregarding the abundantly present hepatocytes. However, dissociation of cultured PCLS is not easy, and many slices have to be pooled, as the cell yield is very low. Bartucci et al. published the only paper where dissociation of cultured PCLS is described. They performed enzymatic digestion of 12 pooled (5 mm) slices to obtain a single cell suspension, after which flow cytometry was used to analyze the uptake of nanoparticles of individual cells [56]. We are also not aware of reports that have carried out single cell RNA sequencing on slices cultures. The latter could give more insight into the cellular characteristics of the slices and the resemblance with their in vivo (fibrotic) counterpart.


ijms-22-07137-t002_Table 2Table 2Recommended minimal read-outs for disease analysis. For every readout one literature reference is given that illustrates the readout for the indicated PCLS application. * Not part of minimal requirements.Liver DiseaseRecommended Read-OutsToxicologyHepatocyte viability by LDH leakage or ATP assay [86]Hepatocyte viability by functionality through Albumin ELISA [73]FibrosisLDH leakage or ATP assay (hepatocyte viability) [71]mRNA levels of fibrosis markers by qRT-PCR [24]Protein levels of fibrosis markers (WB, staining or ELISA) [24]Sirius Red staining for cross-linked collagen [24]Fatty liver diseaseLDH leakage or ATP assay [58]Lipid Staining (e.g., ORO) [58]Triglyceride analysis/quantification [58]qRT-PCR for fibrosis mRNA markers [87]Alcoholic liver diseaseLDH leakage or ATP assay [42]Medium analysis for metabolites [42,88]qRT-PCR for fibrosis mRNA markers [42,85]Electron microscopy [65] *Inflammation/ImmunityELISA on medium for inflammatory markers [48,72]qRT-PCR for inflammatory mRNA markers [89]CholestasisLDH leakage or ATP assay [90,91]qRT-PCR for cholestatic mRNA markers [91]Bile acid determination [91]CancerLDH leakage or ATP assay [21]qRT-PCR for cancer-related mRNA markers [21]H&E staining (tumor morphology) [54]Staining for Ki67 (proliferation) [54]


Bioreactor technologies—Research has already confirmed that a continuous flow (perfusion) and exchange in culture medium improves the lifespan of PCLS culture compared to static or dynamic cultures [18]. However, less than a handful of research papers further focus on the optimization of this bioreactor technology. Van Midwoud et al. describe a microfluidic biochip containing a micro chamber (25 µL) wherein rat slices (100 µm thick, 4 mm diameter) were cultured and perfused (flow rate: 10 µL/min) for 24 h [92]. When comparing this ‘bioreactor’ set-up with the ‘conventional’ culture (i.e., well plate on shaking platform), no significant differences in LDH leakage were observed between both systems. Moreover, similar metabolization rates were seen. In follow-up papers, using the same microfluidic biochip, they show that the methodology can be extended to other organs (intestine) [93] and that the system allows the use of a matrigel-based hydrogel [94]. Although the major advantage of the biochip was that it allowed the control of the medium flow, biochip cultured liver slices (with or without hydrogel) did not show improved characteristics compared to the conventional well plate-cultured discs.

Paish et al. have published the latest advances of applying bioreactor technologies to PCLS cultures in 2018. The research group developed a modified 12-well culture plate (BioR plate) to culture rat and human PCLS, while these plates were rocked on a bioreactor platform at a flow rate of 18.14 µL/s. The bioreactor set-up extended the health, metabolic activity and functional longevity of the PCLS to at least 6 days, and allowed the modelling of active fibrogenesis, by exposing the PCLS to pro-fibrotic factors TGF-β1 and PDGF-ββ [24]. Although this bioreactor set-up for the culture of PCLS has been applied in two other papers [71], bioreactor technologies are not a standard yet in liver slice cultures. This might be due to the size of the PCLS, limiting its applicability to standardized bioreactor systems that have been developed for seeding cell suspension or transfer of small spheroids (i.e., Insphero, Hemoshear Therapeutics, and TissUse).

Nanotechnology—Drug delivery and gene therapy through nanotechnology could be performed in PCLS as well. The main target liver cell types for nanotechnology are Kupffer cells, hepatocytes, or endothelial cells (LSEC). Dragoni et al. confirmed the internalization and active uptake of gold particles in all three cell types [64]. Bartucci et al., on the other hand, demonstrated the nanosafety after long-term (72 h) exposure to nanoparticles [56]. These experimental set-ups demonstrate the applicability of PCLS cultures not only for disease modelling but clinical applications as well.

Virology—PCLS have been proven to be suitable for viral studies, as infection with e.g., the Hepatitis virus have been established. Kartasheva-Ebertz accomplished viral infection of human slices through inoculation with infected supernatant overnight [26]. This offers opportunities for other liver-associated viral studies.

Gene silencing—Although applied to liver studies in general [95,96], no PCLS research papers have been found where gene silencing is performed. However, siRNAs have been successfully applied to other tissue slices, such as lung tissue slices [97] or kidney slices [98]. The fact that gene silencing using siRNAs can be obtained in liver tissue and other tissue slices does not retain the possibility of its use in liver slices.

Transgenic species—The use of transgenic species or even mice with a humanized liver [99] could aid in mechanistic studies or down-stream analysis of (specific) chronic liver disease in a more human-like setting. Although several genetically modified species are available, e.g., for NAFLD [100] or liver cancer [101], only a few PCLS research papers make use of transgenic species. We recommend the use of transgenic species for PCLS research, as this would improve the progression in chronic liver disease research.

## 4. Future Perspectives

Optimized PCLS cultures would allow for the application of novel techniques and culture methods. Co-cultures of liver tissue with other organs, so called ‘body-on-a-chip’ cultures, do already exist [102,103]. One can imagine that culture systems such as the BioR plate [24] that were optimized for liver slices, can allow co-culture of different organ slices in different wells connected with each other by the small channels. The latest single-cell analysis advances, such as single-cell proteomics [104,105], imaging [106], or sequencing [107,108] will also shed more light on how representative the PCLS cultures are for human or rodent chronic liver diseases. Obviously, slice cultures cannot represent every aspect of human chronic liver disease, but it will be important to determine what aspects can or cannot be modelled by PCLS cultures. Specialized microfluidic devices, that facilitate and standardize the tissue processing could aid in the further employment of such single-cell techniques to PCLS cultures [109]. If PCLS cultures could be downscaled to 96- or 384- well formats they would also become suitable for high-throughput assays as well, and automated cell culture devices (e.g., Viaflow [Integra] or Ambr15 [Sartorius]) could be adapted to facilitate this transition.

## 5. Conclusions

In theory, PCLS have all features to excel as an in vitro tool to study chronic liver disease and the use of PCLS contributes to the 3R principle regarding animal use. Therefore, liver slice cultures could constitute a reliable tool for pre-clinical testing. Still the majority of studies carry out drug-induced liver injury (DILI) experiments and can only evaluate the slices the first 72 h. The focus of most read-outs relies on hepatocyte functionality, survival and metabolism while few studies focus on HSCs, Kupffer cells or endothelial cells. As the focus of PCLS research shifts more towards using the model to study chronic liver diseases, more researchers have started to investigate these sinusoidal cell types as well [110,111,112,113,114,115,116].

Overall, this review shows that no major changes have been implemented in the parameters used to produce and culture PCLS in the past 10 years. While some reports, mainly using human slices, show great improvement in the life span and applicability of the cultures, still the majority of the publications does not achieve this level of sophistication. Hitherto no reports clearly explain why they achieve good cultures while still using the same techniques for generating the slices and culturing them as at the start in 1985. Moreover, it is alarming that not all papers accurately report their used PCLS parameters. This could suggest that some liver slice studies possibly might not be carried out in the most ideal conditions and cannot easily be reproduced due to lacking information. We advocate for better documentation of PCLS parameters used in research papers to enhance reproducibility of the experiments and encourage scientist to openly share those parameters that increase the life span and functionality of liver slice cultures. These practices might in the future lead to more stable PCLS cultures that allow not only the modelling of DILI and the procedure-induced fibrosis, but also drug-induced fibrosis, NAFLD, cholestasis, and phospholipidosis.

## Figures and Tables

**Figure 1 ijms-22-07137-f001:**
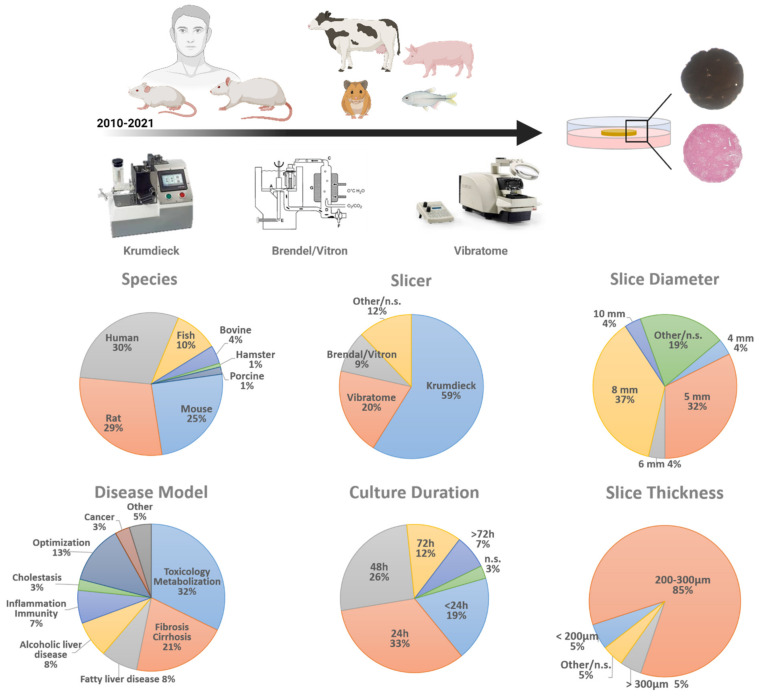
Precision cut liver slices: an overview of parameters in numbers. PCLS parameters of 107 research papers reporting on PCLS in the past 10 years were analyzed. The parameters analyzed are: the origin of liver tissue (species), tissue slicer used, slice diameter, slice thickness, and culture duration. n.s. = not specified.

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
