# Peer review of "Best Practices and Progress in Precision-Cut Liver Slice Cultures"

_ijms, 2021, doi:10.3390/ijms22137137_

Round 1

Reviewer 1 Report

Authors revise most of PCLS articles and compare 107 published articles. It is one of outstanding and tremendous work. I recommended only very few comment and this review should be published as soon as possible.

Minor comments

Please revise the review to remove the spaces and language typos,

Some important review, articles are missing and should be discussed

PMID: 32683516

PMID: 23974980

PMID: 30515676

Author Response

  1. Comment: Authors revise most of PCLS articles and compare 107 published articles. It is one of outstanding and tremendous work. I recommended only very few comment and this review should be published as soon as possible.

Reply: We are happy to hear that our manuscript pleased the reviewer and address the comments below.

  1. Comment: Please revise the review to remove the spaces and language typos.

Reply: Thank you for pointing out the spelling mistakes. Spelling has been checked throughout the entire document. Typo’s and mistakes regarding spaces have been corrected.

  1. Comment: Some important review, articles are missing and should be discussed. PMID: 32683516, PMID: 23974980 PMID: 30515676

(1) Precision‑cut liver slices as an alternative method for long‑term hepatotoxicity studies (Othman 2020).
(2) Recent advances in 2D and 3D in vitro systems using primary hepatocytes, alternative hepatocyte sources and non-parenchymal liver cells and their use in investigating mechanisms of hepatotoxicity, cell signaling and ADME (Godoy 2013)
(3) Precision-cut liver slices: a versatile tool to advance liver research (Palma 2019)

Reply: These reviews indeed give an overview of the latest advances in PCLS research. These have now been included and discussed in the paper (from line 48 on).

Reviewer 2 Report

The authors analysed 107 research papers that use PCLS cultures and summarised the PCLS parameters and most common read-outs. Despite the thorough literature overview, this review is too descriptive and vague, and the added value is unclear. Furthermore, the versatility of PCLS as a model for various liver diseases is not fully reflected.

If the authors state that "the focus of PCLS experiments has shifted from toxicity testing towards studies of chronic liver disease, such as fibrosis" (line 30-31), then the induced fibrogenesis during slicing seems to be less of a disadvantage. In contrast, one might argue that such a spontaneous onset of fibrogenesis (i.e. without the need to add an exogenous inducer) mimics the multifactorial fibrotic liver disease better than mouse models where fibrosis is induced by triggers that are rarely, if ever, seen in human disease (e.g. toxic injury by CCl4).

My suggestion to the authors would be to change the angle of this review and re-write it as a methods paper instead. As authors correctly noticed, the current PCLS protocol was published in 2010 and hasn't changed still, while PCLS cultures found new applications for which changes in parameters/read-outs are needed. In my opinion, such article that aims to help researchers to establish PCLS cultures in their lab (hence, guides them through all the points that have to be considered such as equipment/slice diameter/thickness/costs) or helps researchers to adapt PCLS model for a specific research question (by for example modifying culture conditions: for NASH research, one might want to add fatty acids to the medium, etc) would be of great interest for readers. 

This way, the authors can showcase the variety of applications for which PCLS cultures can be used, not limiting it to toxicity/metabolism studies and fibrosis, but also expand it to the fields of nanotechnology, virology and gene silencing. In addition, the variety of assays and read-outs can be presented not in the terms of "often used" or "not generally applied" but in the light of what PCLS can offer. 

Figure 1 would be interesting in the introduction of such methods paper as well (how PCLS were used in research papers of the last 10 years), and I advise the authors to go beyond it: is there anything we can learn from tissue slices from other organs (in terms of for example, culture conditions or possible applications) that maybe should be considered for PCLS? The authors touched upon the topic of long-term cultures which is really interesting, and it would be nice to expand on that.

What would make the article to stand out is the addition of some kind of "decision tree" that researches can use to select optimal parameters and read-outs for their specific research area/question within liver diseases field. In other words, if a researcher would like to establish PCLS cultures for (for example) studying NASH or long-term drug toxicity, what he/she should take into consideration?

Lastly, a small discussion about where, in the opinion of authors, the future of PCLS cultures lies (any new areas in liver research field? new exciting assays? establishing new hybrid or co-culture systems with the use of PCLS?) would also add value to the article.

The review in its current form lacks scientific significance, however the topic of "best practices" in PCLS is definitely worth of exploring!

Author Response

We thank the reviewer for the valuable feedback. We have taken the comments into account as much as possible for the revised manuscript. For each comment, we replied accordingly and/or made the necessary changes.

  1. Comment: The authors analysed 107 research papers that use PCLS cultures and summarised the PCLS parameters and most common read-outs. Despite the thorough literature overview, this review is too descriptive and vague, and the added value is unclear. Furthermore, the versatility of PCLS as a model for various liver diseases is not fully reflected.

Reply: We are convinced of the potential of PCLS to study various liver disease. In the review, we emphasize that despite the opportunities of PCLS as a model for liver disease, these slices are occasionally used (only 29%!) for modelling chronic liver disease, other than fibrosis, toxicity, compound metabolization or PCLS optimization.

  1. Comment: If the authors state that "the focus of PCLS experiments has shifted from toxicity testing towards studies of chronic liver disease, such as fibrosis" (line 30-31), then the induced fibrogenesis during slicing seems to be less of a disadvantage. In contrast, one might argue that such a spontaneous onset of fibrogenesis (i.e. without the need to add an exogenous inducer) mimics the multifactorial fibrotic liver disease better than mouse models where fibrosis is induced by triggers that are rarely, if ever, seen in human disease (e.g. toxic injury by CCl4).

Reply: We disagree with the reviewer on this aspect and would like to emphasize that the so-called ‘spontaneous’ onset of fibrogenesis does originate from an exogenous inducer (explained line 39-41), namely the slicing of the tissue. We doubt that such fibrotic signature would be representative for human liver disease. The culture/procedure induced fibrogenesis and its inherent advantages and disadvantages are more thoroughly discussed later in the paragraph ‘Disease model’ (line 197-204).

  1. Comment: My suggestion to the authors would be to change the angle of this review and re-write it as a methods paper instead. As authors correctly noticed, the current PCLS protocol was published in 2010 and hasn't changed still, while PCLS cultures found new applications for which changes in parameters/read-outs are needed. In my opinion, such article that aims to help researchers to establish PCLS cultures in their lab (hence, guides them through all the points that have to be considered such as equipment/slice diameter/thickness/costs) or helps researchers to adapt PCLS model for a specific research question (by for example modifying culture conditions: for NASH research, one might want to add fatty acids to the medium, etc) would be of great interest for readers. 

Reply: As the reviewer mentions already, the PCLS protocol published in 2010 has not changed considerable and thus a new methods paper is not really relevant. For each PCLS parameter given in this paper, we highlight the considerations one could/should make. However, for these changes in parameters, no to few documentation was found. Very few comparisons in parameters used are made, thus making it impossible to re-write this review as a methods paper. This review so thus highlights the gap in PCLS research/literature and is also a call for change in how scientist report their PCLS studies.

The reviewer suggests to guide researchers in setting-up PCLS cultures in the lab. As mentioned, we discuss all parameters required for producing PCLS cultures, each with their advantages and disadvantages, and reasoning of choice as best as we could.

We agree with the reviewer that it would be very informative to give an overview of the culture conditions required for the modeling of liver diseases such as NASH. However, sufficient PCLS research studies concerning these topics need to be available in order to do guide scientist through the possible conditions. As mentioned before, the majority of PCLS research papers involves research concerning liver fibrosis and toxicity testing and very few document other liver diseases such as NAFLD or NASH. Moreover, showing that there is lipid loading when there is an excess of free fatty acids added to a slice culture does not mean that there is NASH. In conclusion we cannot give with certainty the best PCLS method to address a certain research question, which is why we wrote this review paper and not a methods paper.

  1. Comment: This way, the authors can showcase the variety of applications for which PCLS cultures can be used, not limiting it to toxicity/metabolism studies and fibrosis, but also expand it to the fields of nanotechnology, virology and gene silencing. In addition, the variety of assays and read-outs can be presented not in the terms of "often used" or "not generally applied" but in the light of what PCLS can offer. 

Reply: We thank the reviewer for pointing us towards these additional applications.  We have now introduced three additional paragraphs on virology, nanotechnology, gene silencing and transgenic species. These additional applications for PCLS research have been implemented (line 465).

  1. Comment: Figure 1 would be interesting in the introduction of such methods paper as well (how PCLS were used in research papers of the last 10 years), and I advise the authors to go beyond it: is there anything we can learn from tissue slices from other organs (in terms of for example, culture conditions or possible applications) that maybe should be considered for PCLS? The authors touched upon the topic of long-term cultures which is really interesting, and it would be nice to expand on that.

Reply: We agree with the reviewer that longevity is an interesting topic and we briefly address this now from line 161 on.

  1. Comment: What would make the article to stand out is the addition of some kind of "decision tree" that researches can use to select optimal parameters and read-outs for their specific research area/question within liver diseases field. In other words, if a researcher would like to establish PCLS cultures for (for example) studying NASH or long-term drug toxicity, what he/she should take into consideration?

Reply: We’ve made an overview of recommended read-outs for each specific research topic/investigated liver disease. This can be found back from line 419 on, the table can be found back from line 436 on.

  1. Comment: Lastly, a small discussion about where, in the opinion of authors, the future of PCLS cultures lies (any new areas in liver research field? new exciting assays? establishing new hybrid or co-culture systems with the use of PCLS?) would also add value to the article.

Reply: Future perspectives have been addressed in the paper (line 491).

Comment: The review in its current form lacks scientific significance, however the topic of "best practices" in PCLS is definitely worth of exploring!

Round 2

Reviewer 2 Report

The authors have addressed the main concerns, and the manuscript is significantly improved. Table 2 is an excellent addition, as well as the future perspectives.